# Transplantation of chicken egg white extract-induced rabbit PBMCs as a treatment for renal ischemia-reperfusion injury in rabbits

**Guang-ping Ruan** [1,2,3]*, **Xiang Yao**[1,2,3], **Qing-keng Lin**[1,2,3], **Zi-an Li**[1,2,3], **Xue-min Cai**[1,2,3], **Rong-qing Pang**[1,2,3], **Xing-hua Pan**[1,2,3]*

1 Kunming Key Laboratory of Stem Cell and Regenerative Medicine, 920th Hospital of the Joint Logistics Support Force of the PLA, Kunming, Yunnan Province, China, 2 Stem Cell and Immune Cell Biomedical Technique Integrated Engineering Laboratory of State and Region, Kunming, Yunnan Province, China, 3 Cell Therapy Technology Transfer Medical Key Laboratory of Yunnan Province, Kunming, Yunnan Province, China

* ruangp@126.com (GR); xinghuapan@aliyun.com (XP)

**Data Availability Statement:** All relevant data are within the manuscript.

**Funding:** This work was supported by grants from Yunnan Science and Technology Plan Project

## Abstract

Ischemia-reperfusion injury is an important contributor to acute kidney injury and a major factor affecting early functional recovery after kidney transplantation. We conducted this experiment to investigate the protective effect of induced multipotent stem cell transplantation on renal ischemia-reperfusion injury. Forty rabbits were divided into four groups of 10 rabbits each. Thirty rabbits were used to establish the renal ischemia-reperfusion injury model, and ten rabbits served as the model group and were not treated. Among the 30 rabbits with renal ischemia-reperfusion injury, 10 rabbits were treated with induced peripheral blood mononuclear cells (PBMCs), and 10 other rabbits were treated with noninduced PBMCs. After three weekly treatments, the serum creatinine levels, urea nitrogen levels and urine protein concentrations were quantified. The kidneys were stained with hematoxylin-eosin (HE), periodic acid-Schiff (PAS) and Masson's trichrome and then sent for commercial metabolomic testing. The kidneys of the rabbits in the model group showed different degrees of pathological changes, and the recovery of renal function was observed in the group treated with induced cells. The results indicate that PBMCs differentiate into multipotent stem cells after induction and exert a therapeutic effect on renal ischemia-reperfusion injury.

## Introduction

Clinical acute kidney injury, which has a very high incidence, might be caused many factors, and ischemia-reperfusion injury is the most common of these factors [1]. The main pathological manifestation of acute kidney injury is renal tubular epithelial damage, and the recovery of renal function depends on renal tubular epithelial cell regeneration. In the event of renal tubular necrosis, blood purification is the only method that can maintain a stable internal bodily environment due to the impairment of kidney function. The development of drugs or interventions that accelerate renal tissue repair during renal tubular epithelial self-repair processes

Major Science and Technology Project (2018ZF007) and the Yunnan Province Applied Basic Research Program Key Project (2018FA041, 2017FA040). National Natural Science Foundation (31970515), 920th Hospital of the PLA Joint Logistics Support Force Inhospital technology plans (2019YGB17, 2019YGA05). The funders had no role in study design, data collection and analysis, decision to publish, or preparation of the manuscript.

**Competing interests:** The authors have declared that no competing interests exist.

will undoubtedly shorten the course of acute kidney injury. Early recovery of a stable internal bodily environment can reduce the complications caused by renal failure and thereby reduce mortality [2].

Renal ischemia-reperfusion injury is clinically common in the context of emergency shock recovery, kidney surgery and kidney transplantation and is thus an important factor affecting renal function [3]. The manifestations of renal ischemia-reperfusion injury include acute tubular epithelial cell injury, tubular peripheral microvascular damage, inflammation and leukocyte infiltration. Mesenchymal stem cells are effective for the treatment of ischemia-related organ dysfunction, but the exact mechanism through which they improve renal function remains unclear [4]. In this study, we investigated the protective effect of induced multipotent stem cell transplantation on renal ischemia-reperfusion injury in rabbits, and our results provide experimental evidence and ideas for clinical treatment.

Induced multipotent stem cell regeneration of renal function represents a new method for the treatment of acute tubular necrosis. In our laboratory, induced multipotent stem cells are generated by the treatment of peripheral blood mononuclear cells (PBMCs) from rabbit peripheral blood with a homemade egg white extract to reverse their differentiation [5]. This study is the first to induce PBMCs with chicken egg white extract and use the induced PBMCs for the treatment of rabbit renal ischemia-reperfusion injury. Previous studies have shown that extracts of mammalian oocytes [6] and Xenopus oocytes [7] have the potential to reprogram cells. The identification of egg extracts with the ability to maintain and enhance the survival and differentiation of cells will be widely useful in cellular biology research. Many studies have shown that animal egg extracts are able to induce the reprogramming of somatic cells [6, 8]. The chicken egg yolk is the largest egg cell, the yolk membrane comprises the cell membrane, and the egg white and eggshell, which have nutritional and protective roles, are formed by oviduct secretions. Therefore, chicken egg white extract has the capacity to induce stemness in PBMCs. In our previous research, we found that chicken egg white extract can induce the reverse differentiation of somatic cells into multipotent stem cells [5, 9].

As determined by quantitative PCR, the induction of PBMCs with the chicken egg white extract significantly increased the relative expression of the multipotency-related genes *OCT4*, *NANOG* and *SOX2* and significantly decreased that of the somatic cell gene *LMNA*, which indicates that PBMCs differentiate into multipotent stem cells [9]. Further experiments are needed to identify the key molecules in the chicken egg white extract to further improve the induction efficiency and promote this method.

Mesenchymal stem cells with multidirectional differentiation potential have been differentiated into renal tubular epithelial cells and effectively promote the restoration of renal function in injured kidneys [4, 10–13]. In the present study, we assessed whether induced PBMCs function as mesenchymal stem cells. To this end, we divided 40 rabbits into four groups: a normal control group, a model control group, an induced cell treatment group and a noninduced cell treatment group. After successful establishment of the model, we compared the efficacies of the treatments and ultimately revealed that the induced PBMCs function as multipotent stem cells and participate in the repair of renal injury.

## Materials and methods

### 1. Preparation of the rabbit model of renal ischemia-reperfusion injury

According to the inclusion and exclusion criteria, 40 wild-type Japanese white rabbits were numbered according to their body mass and randomly allocated to a group. The rabbits were purchased from Kunming Chushang Technology Co., Ltd., and the license number was SCXK (Dian) K2018-0001. The rabbits were anesthetized with intravenous injections of 3% sodium

pentobarbital (Merck, Darmstadt, Germany) at a dose of 1 ml per kg of body weight. The abdominal skin of the rabbits was prepared, and an incision was made along the midline of the abdomen. The skin, subcutaneous tissue, and the peritoneum were separated layer by layer, and an incision was then made in the abdominal cavity. The renal pedicle was located, and the bilateral renal pedicle was rapidly clipped with a noninvasive arterial clip to restrict the blood flow for 1 hour. The artery clip was then released to restore the blood flow and allow reperfusion, which resulted in ischemia-reperfusion injury. The model was established in 30 rabbits, and the other 10 rabbits were not subjected to the operation and served as the normal control group. Among the rabbits with ischemia-reperfusion injury, 10 were used as the model control group, 10 other rabbits were injected with induced PBMCs, and the remaining 10 were injected with noninduced PBMCs. The cells were transfused on the first day after model establishment and then once a week for 3 consecutive weeks. All experimental protocols were approved by the Experimental Animal Ethics Committee of the 920th Hospital of the Joint Logistics Support Force of the People's Liberation Army (approved number: 2019-002-01). No human subjects were included in this study, and thus, informed consent is not applicable. The DOI link of experimental steps is dx.doi.org/10.17504/protocols.io.bpyrmpv6.

## 2. Rabbit PBMC isolation, culture, induction and labeling

Twenty milliliters of peripheral blood was harvested from each of three rabbits, and the experiment was performed more than three times to ensure repeatability. PBMCs were isolated using a separation fluid (Haoyang Biological Products Technology Co., Ltd., Tianjin, China). After two washes, the cells were cultured in culture flasks. The cells in one flask were cultured in DMEM-F12 medium supplemented with 10% fetal bovine serum, and the cells in the other flask were cultured with 50% egg white extract in culture medium. As shown in our previous study, cells grown in 50% chicken egg white extract in culture medium will differentiate into multipotent stem cells and might be used for the treatment of diseases [9]. Thus, we used 50% chicken egg white extract-induced rabbit PBMCs as a treatment for kidney injury in the rabbit model. Three days after induction, CFSE-labeled cells were transfused into the rabbits belonging to the two treatment groups; CFSE fluoresces yellowish-green. The cells were centrifuged, and the supernatant was discarded. The cell pellet was suspended in 4 ml of PBS, and 10 μl of 10 mM CFSE (from Abcam, ab145291, Cambridge, UK) was added. The mixture was incubated at 37°C for 10 minutes, and 3 ml of medium was added to terminate the labeling reaction. The cells were centrifuged, and the supernatant was removed. The cells were suspended in saline, and each rabbit received a transfusion of $2 \times 10^6$ cells. The cell survival rate was greater than 95%. The PBMCs were injected via the intravenous route. We observed that the fluorescence lasted for more than 1 month after the cells were labeled with CFSE, and because the experimental period was less than a month, the fluorescent cells could be traced.

## 3. Identification of noninduced and induced PBMCs

**3.1. Quantitative PCR-based detection of the relative expression of pluripotency-related and somatic cell genes.** Three days after induction, RNA was extracted from the PBMCs, reverse transcribed, and subjected to quantitative PCR to determine the relative levels and fold changes in the expression of the pluripotency-related genes *OCT4*, *NANOG*, and *SOX2* and the somatic cell gene *LMNA*. The primer sequences and product lengths are shown in Table 1.

**3.2. Flow cytometry.** Three days after induction, the PBMCs were labeled with SSEA-4-PE, OCT4-PE and NANOG-PE, and the changes in the expression of pluripotency markers were detected.

**Table 1. Primer sequences used in the quantitative PCR-based detection of pluripotency-related and somatic genes and lengths of the products.**

| Gene | Primer sequences | Product length (bp) |
|---|---|---|
| OCT4 | F: 5′-AAGGAGAAGCTGGAGCAAACC-3′ | 164 |
| | R: 5′-CTGAACACCTTTCCAAAGAGAACCC-3′ | |
| NANOG | F: 5′-TCAGCCTTCAGCAGATGCAA-3′ | 150 |
| | R: 5′-GGCACCCCTGAGTCACAC-3′ | |
| SOX2 | F: 5′-AACGCCTTCATGGTATGGTC-3′ | 253 |
| | R: 5′-CTCCGGGAAGCGTGTACTTA-3′ | |
| LMNA | F: 5′-CTTGCTGACTTACCGCTTCC-3′ | 253 |
| | R: 5′-CAGGTCATCTCCATCCTCGT-3′ | |
| GAPDH | F: 5′-CGAGACACGATGGTGAAGGT-3′ | 139 |
| | R: 5′-TGTAGACCATGTAGTGGAGGTC-3′ | |

SSEA-4 is a surface-expressed antigen. The PBMCs were centrifuged, and the supernatant was discarded [9]. The pellet was suspended in 50 μl of PBS, and 20 μl of an isotype control antibody or SSEA-4-PE was added to the wells. The mixtures were incubated for 1 hour at room temperature in the dark, and the cells were then washed once with PBS, fixed with 4% paraformaldehyde in PBS, and detected using a flow cytometer (Manufacturer: Becton Dickinson; model: BD LSRFortessa; software version: BD FACSDiva Software v8.0.1, USA). All the antibodies were used in strict accordance with the manufacturer's instructions to avoid non-specific binding.

**3.3. Immunohistochemical detection.** Noninduced and induced PBMCs were subjected to immunohistochemistry analysis. Specifically, noninduced and induced PBMCs were smeared on a slide, dried, and fixed with 4% paraformaldehyde for 10 minutes. The cells were stained with an Abcam immunohistochemistry kit using primary antibodies against OCT4 and NANOG at 1:50 dilution. The procedures were performed according to the manufacturer's instructions. DAB was used to develop the staining, and images were captured.

**3.4. Western blot detection.** The western blot detection of noninduced and induced PBMCs was performed as follows. Proteins were extracted from noninduced and induced PBMCs, electrophoresed, and transferred to a membrane. The membrane was blocked for 1 hour and then incubated with a 1:500 dilution of an anti-OCT4 primary antibody for 1 hour at 37˚C with shaking. The membrane was subjected to three 5-minute washes with TTBS and then incubated with a 1:500 dilution of the secondary antibody for 1 hour at 37˚C with shaking. The membrane was subjected to three 5-minute washes with TTBS. Enhanced chemiluminescence (ECL) was used to detect the proteins, and images were captured using a chemiluminescence imaging camera (Tanon 5200, Shanghai).

**3.5. Quantitative PCR detection of relative changes in the telomere length.** The relative changes in the telomere lengths in noninduced and induced PBMCs were detected as follows. Three days after induction, RNA was extracted from the PBMCs, reverse transcribed, and subjected to quantitative PCR to determine the relative changes in the telomere length. The telomere and internal control 36B4 primer sequences are shown in Table 2.

**Table 2. Primer sequences for telomeres and the internal control 36B4.**

| Genes | Primer sequences |
|---|---|
| Telomere | Forward: 5′-CGGTTTGTTTGGGTTTGGGTTTGGGTTTGGGTTTGGGTT-3′ |
| | Reverse: 5′-GGCTTGCCTTACCCTTACCCTTACCCTTACCCTTACCCT-3′ |
| 36B4 | Forward: 5′-CAGCAAGTGGGAAGGTGTAAATCC-3′ |
| | Reverse: 5′-CCCATTCTATCATCAACGGGTACAA-3′ |

## 4. Kidney function test

After three transplantations of the labeled cells, blood samples were collected from the ear veins of rabbits in the four groups. Serum was separated and sent to a laboratory to examine the changes in the creatinine and urea nitrogen contents.

## 5. Changes in the urinary protein content

After three transplantations of the labeled cells, urine samples were collected from the four groups of rabbits, and the urine protein contents were measured using the BCA method.

## 6. Observation of labeled cells in frozen sections

After three transplantations of the labeled cells, three rabbits from each group were euthanized through the introduction of an air embolism. The renal distributions of the labeled cells in frozen kidney sections were observed. The kidney tissue collected was partly frozen to observe fluorescent cells, and the histology was partially evaluated.

## 7. Histopathological assessment of the kidneys

Kidney tissue was used for HE staining and immunohistochemistry analysis of the fibrosis-related factor TGF-β, Masson's trichrome staining and PAS staining. Changes in the renal tissues from each group were observed through HE staining. The degree of renal fibrosis was detected by immunohistochemistry. The extent of fibrosis in the kidneys from each group was detected by Masson's trichrome staining. The thickening of the renal basement membrane was detected by PAS staining. The kidney tissues of the treatment group were frozen, sectioned, incubated with the primary antibody aquaporin-1 (from Abcam, ab9566, Cambridge, UK) specific for the proximal tubules, incubated with a red fluorescence-labeled secondary antibody (from Abcam, ab150115, Cambridge, UK), and counterstained with DAPI, and the distribution of red, green, and blue fluorescence was observed under a fluorescence microscope.

## 8. Metabolomics analysis of the kidney

After three transplantations of the labeled cells, five rabbits from each of the four groups were euthanized via introduction of an air embolism. The kidney tissues were rapidly removed, placed in liquid nitrogen, refrozen at -80°C, and then sent to a company on dry ice for metabolomics analysis. Five biological replicates of rabbit kidney tissue samples from the four groups were analyzed, and the 20 samples were then analyzed by LC-QTOFMS. The ionization source of the LC-QTOFMS platform was electrospray ionization. Two ionization modes, positive ion mode (POS) and negative ion mode (NEG), were used for the detection of metabolites and increase the coverage rate and thus the detection rate.

## 9. Statistical analysis

All statistical analyses were performed using SPSS 21.0 statistical software. The measurement results are expressed as the means±standard deviations. Four-group comparisons and the above results were analyzed by one-way analysis of variance (one-way ANOVA), and $p < 0.05$ was considered statistically significant. Post hoc comparisons between individual groups after ANOVA were needed for the assessment of significant differences between specific groups. Pairwise comparisons between groups were performed using the LSD and SNK methods.

## Results

### 1. Successful labeling of PBMCs with CFSE

Carboxyfluorescein diacetate and succinimidyl ester (CFSE)-labeled PBMCs exhibited yellow-green fluorescence under a fluorescence microscope, as shown in Fig 1B. After counterstaining with DAPI, all the cells showed blue fluorescence, as shown in Fig 1A, and an analysis of the overlap in the fluorescence staining showed that the cells were successfully labeled, as shown in Fig 1C.

### 2. Identification of PBMCs after induction

**2.1. Relative expression levels of multipotency-related and somatic cell genes in noninduced and induced PBMCs.** The expression of *NANOG*, *OCT4* and *SOX2* was significantly increased in PBMCs stimulated with chicken egg white extract, and the expression of the somatic cell gene *LMNA* was decreased (Fig 1D), which indicated that the cells differentiated into multipotent cells. A statistical analysis showed that the two groups were significantly different (n = 3, p = 0.003). The cells were induced three times, and three biological replicates were included in the study. The difference in the somatic cell gene *LMNA* between the two groups was not statistically significant.

**2.2. The pluripotency factor of induced PBMCs was significantly higher than that of noninduced PBMCs.** Among the noninduced cells, 0.081% were positive for OCT4-PE, whereas 99.3% of the induced cells were positive for OCT4-PE. The percentages of SSEA-4-PE-positive noninduced and induced cells were 1.08% and 16.5%, respectively. Moreover, 0.495% of the noninduced cells were positive for NANOG-PE, whereas 95.8% of the induced cells were found to be NANOG-PE-positive (Fig 1E–1M). The percentages of isotype control cells positive for OCT4-PE, SSEA-4-PE and NANOG-PE were 0.546%, 0.401% and 0.249%, respectively.

**2.3. Immunohistochemical analyses yielded positive results for the induced PBMCs.** Immunohistochemical analyses of OCT4 and NANOG showed that induced PBMCs, but not noninduced PBMCs, expressed these markers (Fig 2A–2D). Fig 2A and 2C show uninduced PBMCs, and Fig 2B and 2D show induced PBMCs. In addition, OCT4 expression is shown in Fig 2A and 2B, and Fig 2C and 2D show the expression of NANOG.

**2.4. Western blot analyses yielded positive results for induced PBMCs.** According to the Western blot results, OCT4 was expressed in induced PBMCs but not in noninduced PBMCs (Fig 2E).

**2.5. The relative telomere lengths were significantly longer in induced PBMCs.** Based on the quantitative PCR results, the telomeres of induced PBMCs (1.83807±0.84756) were significantly longer than those of noninduced PBMCs (1±0.08307) (Fig 2F, means±standard deviations, n = 5, p = 0.013), which indicated that the cells differentiated into young stem cells. The relative telomere lengths actually increased after dedifferentiation.

### 3. The serum urea nitrogen and creatinine levels were decreased in the induced cell treatment group

The model control group had a urea nitrogen content of 22.1 mmol/l and a creatinine content of 452 μmol/l. After three rounds of transplantation with induced cells, the urea nitrogen content was 7 mmol/l, and the creatinine content was 74 μmol/l; these values were similar to the normal levels. The levels in the noninduced cell treatment group remained elevated (Fig 2G). A statistical analysis showed statistically significant differences among the results of the four groups (p = 0.031).

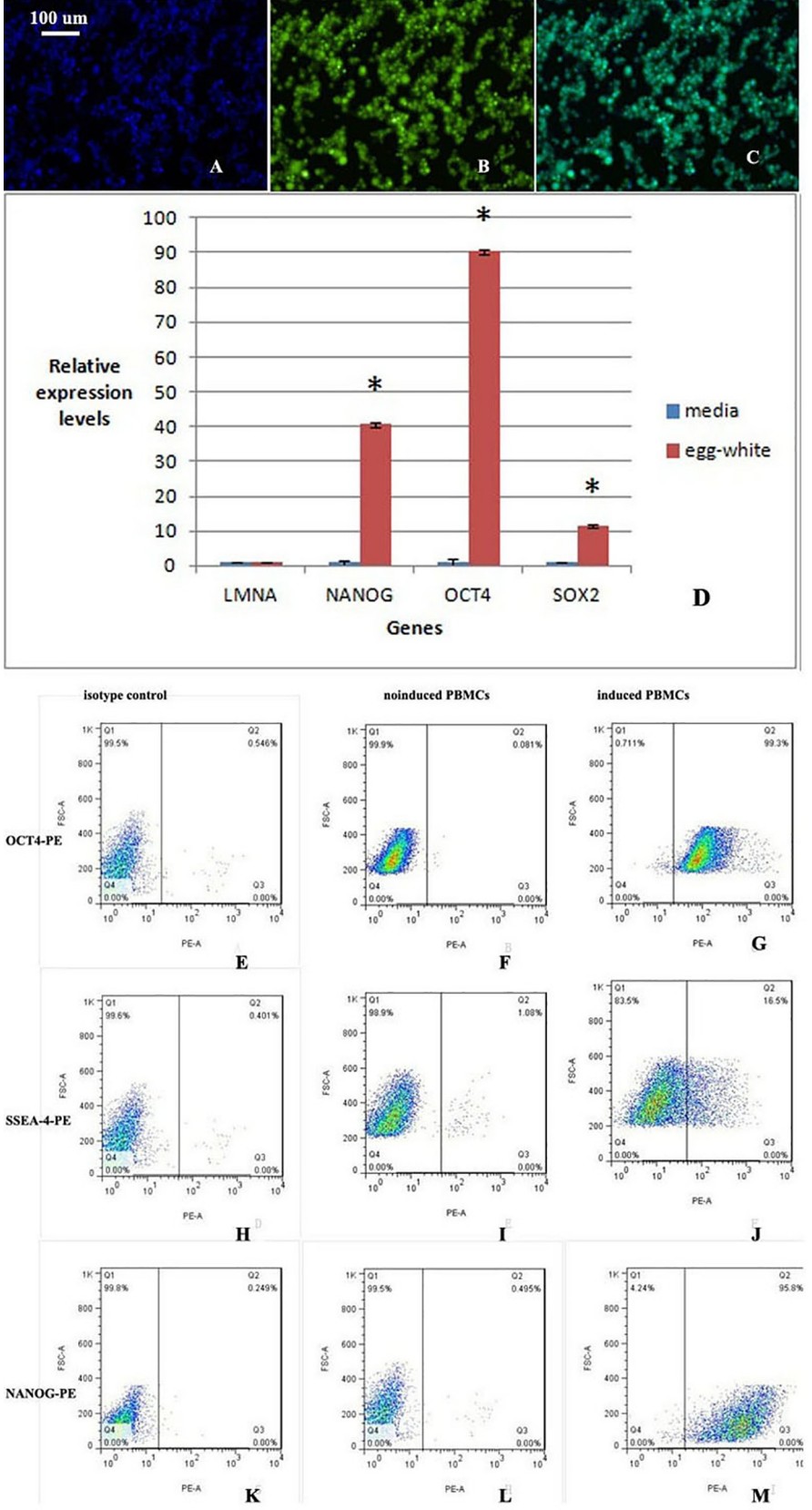

**Fig 1. CFSE-labeled PBMCs display yellow-green fluorescence.** The background is black, and the cells show yellow-green fluorescence. A. DAPI shows blue fluorescence. B. CFSE-labeled PBMCs display yellow-green fluorescence. C. Overlap between blue fluorescence and yellow-green fluorescence. All the cells are labeled with yellow-green

fluorescence. D. The expression of the pluripotency-related genes *NANOG*, *OCT4*, and *SOX2* was significantly increased after induction, whereas the expression of the somatic cell gene *LMNA* was decreased after induction. A statistical analysis showed that the two groups were significantly different (n = 3, p = 0.003). The difference in the somatic cell gene *LMNA* between the two groups was not statistically significant. E-M. Flow cytometry analyses of noninduced and induced PBMCs. E, F and G: OCT4-PE; H, I and J: SSEA-4-PE; and K, L and M: NANOG-PE. E, H and K: isotype control; F, I and L: noninduced PBMCs; and G, J and M: induced PBMCs. After induction, the proportion of cells positive for multipotency-related factors was significantly increased.

## 4. The urinary protein concentrations were decreased in the induced group

The urinary protein concentration in the model control group was 8.17 mg/ml. After the three treatments, the urinary protein concentration in the induced group was 4.35 mg/ml, which was similar to the normal level. However, the urinary protein content of the noninduced group remained high at 7.96 mg/ml (Fig 2H). A statistical analysis showed statistically significant differences among the results obtained from the four groups (p = 0.001).

## 5. Labeled cells were detected in the induced group

As shown in Fig 3A–3D, many fluorescent cells were distributed in the kidneys of the induced cell treatment group, whereas the kidney tissues from the other three groups did not display any fluorescent cells. A possible explanation is that induced cells are transported to the injured kidney to repair the damage. The distribution of fluorescently labeled induced PBMCs in frozen kidney sections suggested that these cells were involved in repairing the injured kidney.

## 6. The structure of the kidney of the induced group exhibited a normal phenotype

Based on the hematoxylin-eosin (HE) staining results, the structure of the kidneys of the rabbits in the model control group was damaged. After treatment with the induced cells, the renal tissue structure exhibited a normal phenotype. In contrast, the damage to the kidney tissue persisted in the noninduced cell treatment group (Fig 3E–3H). The acute tubular necrosis (ATN) score is shown in Fig 3I.

## 7. The IOD of the induced cell groups was significantly reduced

Image-Pro Plus 6.0 (Media Cybernetics, Inc., Rockville, MD, USA) software was used to perform an immunohistochemical analysis of the cumulative optical density (IOD). For each group, at least six 200× magnification fields were randomly selected from each section, and images were captured. We attempted to view the entire field of vision to ensure that every photograph had the same background. Image-Pro Plus 6.0 software was used to select the same brown color as a uniform standard for judging the positive staining in all the images. Each image was analyzed to determine the IOD of positive staining. The IOD was significantly increased in the model control group, whereas the IODs of the normal control and induced cell groups were reduced and significantly reduced, respectively. The IOD of the noninduced cell treatment group was not markedly reduced compared with that of the model control group (Fig 3J). A statistical analysis showed significant differences among the results of the four groups (p = 0.001).

## 8. Fibrosis was improved in the induced group

Substantial collagen fiber deposition was observed in the model control group, and serious fibrosis occurred. This fibrosis was improved or eliminated in the induced cell treatment

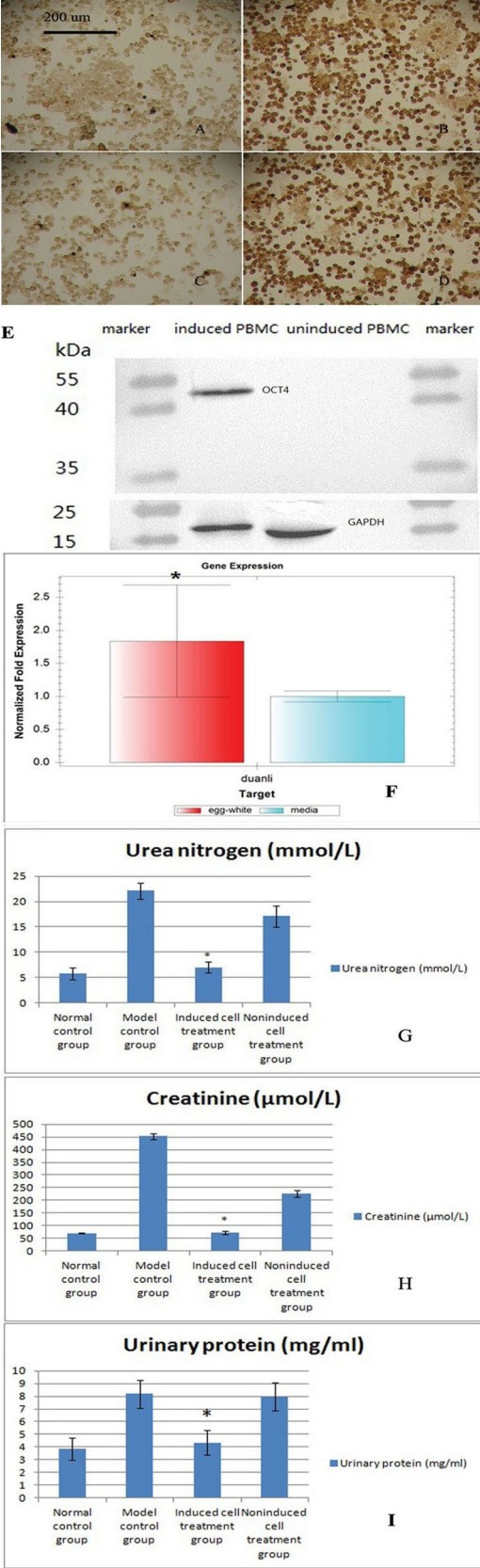

**Fig 2. Immunohistochemical analysis of noninduced and induced PBMCs.** A and C show noninduced PBMCs. B and D show induced PBMCs. The primary antibody used to obtain the results shown in A and B was OCT4, and that used to obtain the results displayed in C and D was NANOG. E. Western blot analyses of noninduced and induced

PBMCs. A primary antibody against OCT4 was used and detected by ECL. The results showed that OCT4 is expressed in induced PBMCs but not in noninduced PBMCs. The internal reference was GAPDH. F. Quantitative PCR analysis of the relative telomere length. The relative telomere length was significantly increased in PMBCs induced with the egg white extract, which indicated that the cells became younger (mean±standard deviation, n = 5, *p = 0.013). G. Serum urea nitrogen levels in the four groups after administration of the three treatments (mean±standard deviation, n = 10). A statistical analysis showed significant differences among the results of the four groups (p = 0.031). H. Serum creatinine levels in the four groups after administration of the three treatments (mean±standard deviation, n = 10). A statistical analysis showed significant differences among the results of the four groups (p = 0.041). I. Quantitative analysis of the urinary protein concentrations in the four groups after administration of the three treatments (mean ± standard deviation, n = 10). A statistical analysis showed significant differences among the results of the four groups (p = 0.001).

group, whereas severe fibrosis was still observed in the noninduced cell-treated group (Fig 4A–4D).

## 9. Thickening of the basement membrane was not observed in the induced group

The thicknesses of the glomerular basement membrane, basilar membrane of the renal capsule and tubulointerstitial membrane were significantly increased in the model control group. In the induced cell treatment group, the basement membrane did not display any significant thickening, whereas thicker basement membranes were still observed in the noninduced cell treatment group (Fig 4E–4H). The basement membrane thickness scores are shown in Fig 4I.

## 10. Transplanted cells differentiated into tubular epithelial cells

The immunofluorescence results showed that the transplanted cells exhibited green and red fluorescence at the same time, which indicated that the transplanted cells were differentiated into tubular epithelial cells (TECs) (Fig 4J).

## 11. The identified substances were increased in the model group and decreased after treatment

In the negative ion mode, the two substances that displayed significantly increased levels in the model control group were 2'-deoxy-D-ribose and N-acetylglucosamine 1-phosphate (Fig 5A). The levels of both of these compounds were significantly reduced after induced cell therapy compared with the levels found in the model control group (Fig 5A). In the positive ion mode, the levels of three substances (D-pinitol, lysyl-glycine and glutamyl-asparagine) were significantly increased in the model control group (Fig 5B), and the levels of these three substances were significantly reduced after induced cell therapy compared with those found in the model control group (Fig 5B).

In the negative ion mode, the levels of the two identified substances were increased in the model group and decreased after treatment, which indicated a meaningful correlation between the levels of the two substances.

In the positive ion mode, the levels of the three identified substances were increased in the model control group and decreased after treatment, which indicated meaningful correlations among the levels of these three substances.

## 11. Significant changes in the pyrimidine metabolism and phenylalanine, tyrosine and tryptophan biosynthesis pathways were observed

In the negative ion mode, a significant change in the pyrimidine metabolism pathway was detected between the model control group and the normal control group (Fig 6A). In addition,

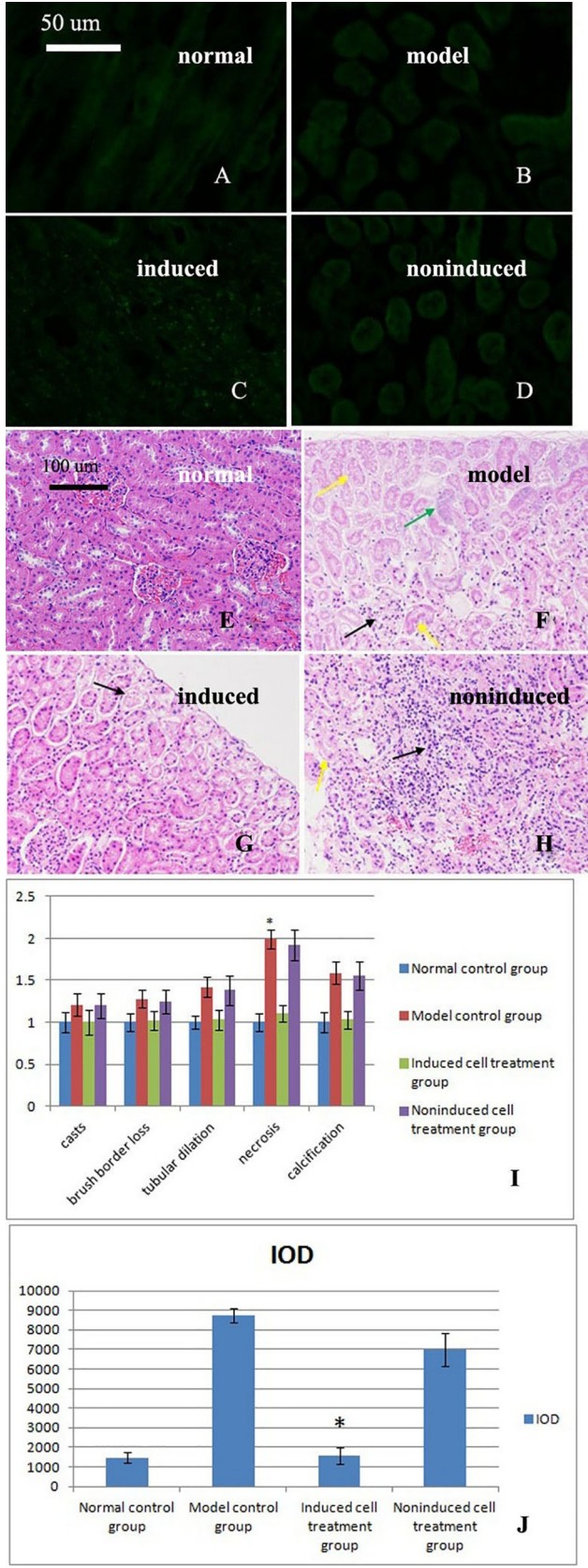

**Fig 3. Observations of labeled cells in frozen kidney sections.** A. No labeled cells were observed in the normal control group. B. No labeled cells were observed in the model control group. C. Labeled cells were observed in the induced cell treatment group. D. No labeled cells were observed in the noninduced cell treatment group. E-H. HE staining of kidneys from the four groups after administration of the three treatments revealed structural changes. E. A normal kidney structure was observed in the normal control group. F. Large amounts of renal tubular necrosis at the edge of the renal cortex and disappearance of the epithelial cell nucleus, as indicated by the yellow arrow, were observed in the model control group. Some of the tubules exhibited slight calcification, as indicated by the green arrows. The numbers of mesangial cells were decreased, and telangiectasia appeared, as indicated by the black arrows. G. A normal kidney structure was observed in the induced cell treatment group. A small amount of renal tubular necrosis was observed at the edge of the renal cortex. Epithelial cell nuclei disappeared, as shown by the black arrow. H. Focal mononuclear cell infiltrates were observed in the noninduced cell treatment group, as indicated by the black arrow. The number of renal tubular epithelial cell nuclei was decreased, and cellular degeneration, cell body swelling, and light cytoplasmic staining were observed, as indicated by the yellow arrows. I. Acute tubular necrosis (ATN) score (mean±standard deviation, n = 3). A statistical analysis showed significant differences among the results of the four groups (p = 0.032). * indicates p<0.05 compared with the induced group. J: TGF-β immunohistochemical analysis (mean±standard deviation, n = 3). A statistical analysis showed significant differences among the results of the four groups (p = 0.022). * indicates p<0.05 compared with the model group.

significant changes in this pathway were observed in the induced cell treatment group compared with the model control group (Fig 6B), which indicated that the pyrimidine metabolism pathway is a significantly altered pathway. In the positive ion mode analysis, the phenylalanine, tyrosine and tryptophan biosynthesis pathways were significantly altered in the model control group compared with the normal control group (Fig 6C), and these pathways were significantly altered in the induced cell treatment group compared with the model control group (Fig 6D), which indicated that the phenylalanine, tyrosine and tryptophan biosynthesis pathways are meaningful pathways. In Fig 6, a darker bubble color and a larger volume indicates a more significant difference. As shown by the arrows in Fig 6A and 6B, the difference in the pyrimidine metabolism pathway is significant, and as indicated by the arrows in Fig 6C and 6D, the differences in phenylalanine, tyrosine and tryptophan biosynthesis pathways are significant.

## Discussion

Acute kidney injury caused by ischemia-reperfusion is a common clinical disease, and the mortality rate of patients with acute kidney injury is approximately 30% to 50% due to limited treatment measures [14]. During ischemia, the activation of enzymes induces cytoskeletal destruction, cell membrane damage, and DNA degradation, which eventually leads to cell necrosis and apoptosis [15]. Ischemia-reperfusion also activates complement proteins, cytokines, and chemokines, among other molecules, and the mechanisms underlying ischemia-reperfusion injury are of substantial importance [16]. Additionally, transplanted stem cells promote the migration and recruitment of residual renal stem cells to the injury site and induce their differentiation into TECs, but the exact mechanism is unclear [17].

The mortality of patients with ischemic-reperfusion injury is high due to the lack of an effective clinical treatment [18]. The condition requires long-term or lifelong renal replacement therapy or renal transplantation [19]. The pathophysiological mechanism of ischemia-reperfusion injury is very complicated and results from numerous interactions among inflammatory cells, vascular endothelial cells and cytokines [20]. In recent years, mesenchymal stem cells have become the focus of research on ischemia-reperfusion injury [21] and have been shown to improve ischemia-reperfusion-induced renal injury. As shown in studies using ischemia-reperfusion models, mesenchymal stem cells play an important role in regulating immunity, participating in vascular reconstruction [22] and repairing the renal microenvironment by secreting paracrine antiapoptotic factors, mitogenic factors and angiogenic factors. A recent study using stem cells of different origins revealed that induced pluripotent stem cells

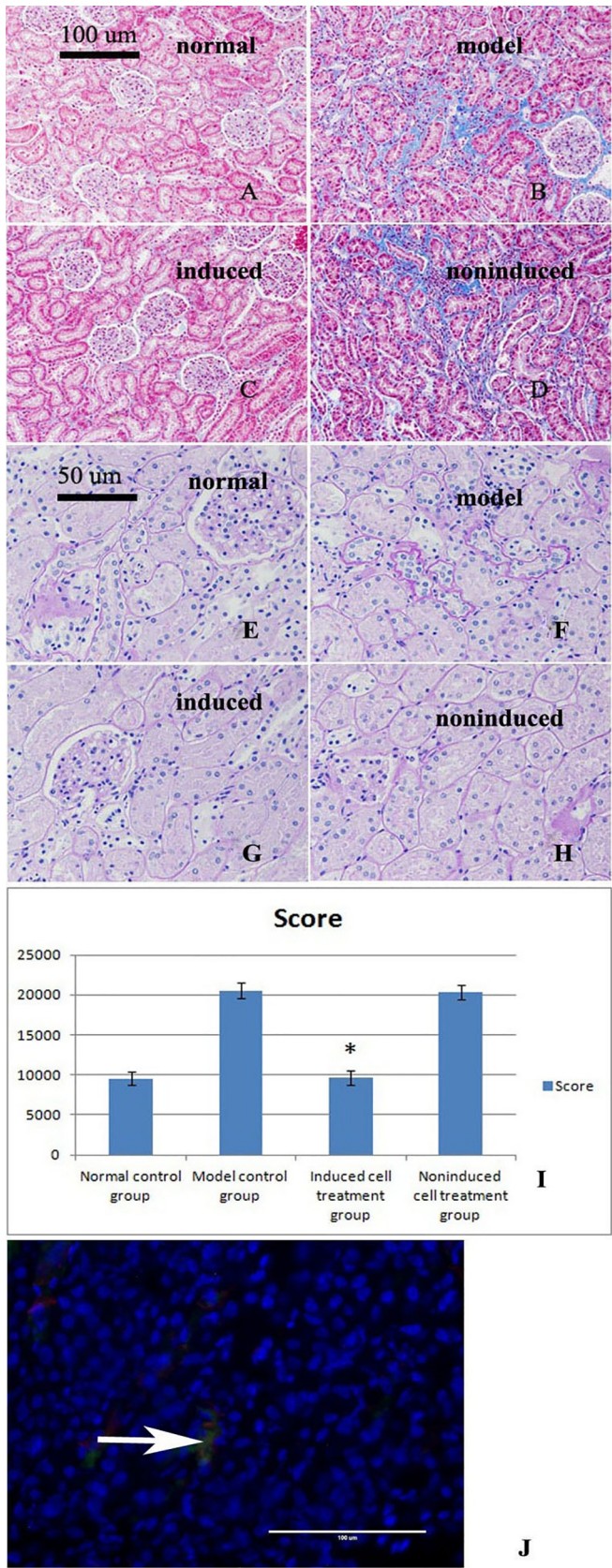

**Fig 4. Masson's trichrome staining of kidney sections from the four groups after the three treatments.** A. The normal control group showed no obvious collagen fiber deposition or fibrosis. B. The model control group showed a large number of collagen fibers, hyperplasia, and severe fibrosis. C. The treatment with the induced cells improved or eliminated the fibrosis observed in the model control group. D. In the noninduced cell-treated group, a large number of collagen fibers, hyperplasia, and severe fibrosis were observed. E-G. Renal PAS staining of the four groups after the three treatments. E. Significant thickening of the basement membrane was not observed in the normal control group. F. Significant thickening of the glomerular basement membrane, basilar membrane of the renal capsule, and tubular basement membrane was observed in the model control group. G. After treatment, significant thickening of the basement membrane was not observed in the induced cell treatment group. H. A thicker basement membrane was observed in the noninduced cell treatment group. I. Basement membrane thickness score (mean±standard deviation, n = 3). A statistical analysis showed significant differences among the results of the four groups (p = 0.024). * indicates p<0.05 compared with the model group. J. Renal tissue immunofluorescence results. The immunofluorescence results showed that the transplanted cells exhibited green and red fluorescence at the same time, which indicated that the transplanted cells had differentiated into tubular epithelial cells (TECs).

exhibit high potential for clinical applications. These cells can migrate to damaged sites, produce large amounts of anti-inflammatory cytokines and growth factors, and exhibit immunomodulatory properties that endow them with greater potential and more advantages than

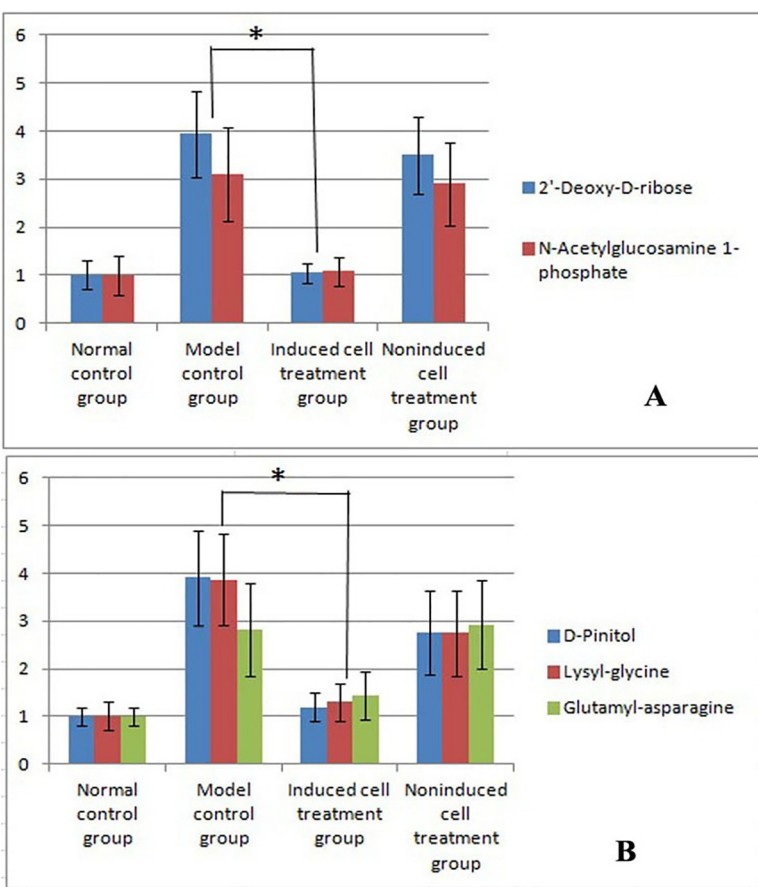

**Fig 5. Results of the metabolomics analysis.** A: The results obtained in the negative ion mode showed significantly increased levels of two substances in the model control group compared with the normal control group (mean ±standard deviation, n = 5). B: In the positive ion mode, the levels of three substances were significantly increased in the model control group compared with the normal control group (mean±standard deviation, n = 5). * indicates p = 0.035 compared with the model group. In the positive ion mode, the levels of three substances were significantly decreased in the induced cell treatment group compared with the model control group (mean±standard deviation, n = 5). * indicates p = 0.041 compared with the model group.

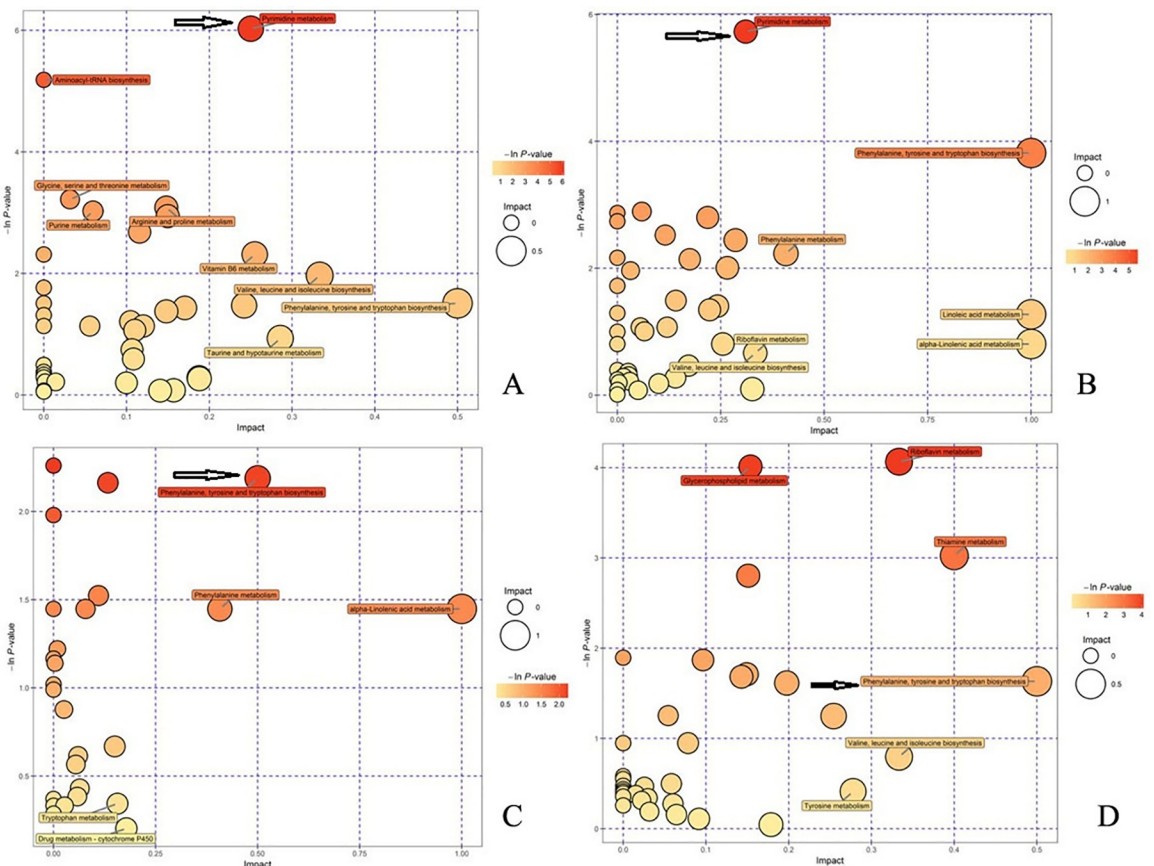

**Fig 6. Analysis of metabolic pathways in the four groups after treatment.** A. In the negative ion mode, the model control group displayed significant changes in the pyrimidine metabolism pathway compared with the normal control group. B. In the negative ion mode, significant changes in the pyrimidine metabolism pathway were also observed in the induced cell treatment group compared with the model control group. C. In the positive ion mode, significant changes in the phenylalanine, tyrosine and tryptophan biosynthesis pathways were detected in the model control group compared with the normal control group. D. In the positive ion mode, the phenylalanine, tyrosine and tryptophan biosynthesis pathways also exhibited significant changes in the induced cell treatment group compared with the model control group. In Fig 6, a darker bubble color and a larger volume indicate a more significant difference. As shown by the arrows in Fig 6A and 6B, the difference in the pyrimidine metabolism pathway is significant, and as indicated by the arrows in Fig 6C and 6D, the differences in the phenylalanine, tyrosine and tryptophan biosynthesis pathways are significant.

other types of stem cells [22]. We believe that the induced cells are more likely to be mesenchymal stem cells because they are more mature than induced pluripotent stem cells.

After induced pluripotent stem cell transplantation, molecules expressed on the surface of the induced pluripotent stem cells interact with T cells to regulate their biological activity. T cells can evade the immune system following damage and reduce the intensity of the immune response in the surrounding tissue, and these cells thus play a role in protecting the function of damaged tissue to some extent [23]. We successfully established a rabbit model of renal interstitial fibrosis and demonstrated that the transplantation of induced autologous stem cells can repair kidney damage within 8 weeks [24]. Stem cells provide an advantageous microenvironment for the repair of renal tubular epithelial damage [24]. In this experiment, after the administration of three consecutive weekly treatments, the serum creatinine and urea nitrogen levels in the induced cell treatment group were restored to the same level observed in the normal control group, whereas the levels in the noninduced cell treatment and model control groups were significantly elevated.

As shown by Haynesworth et al. [25], mesenchymal stem cells secrete a variety of growth factors, colony-stimulating factors, adhesion molecules, and interleukins (IL-6, IL-7, IL-8, IL-11, IL-14, and IL-15), which promote the mitosis of renal TECs and tissue repair and thereby inhibit the elevation of creatinine and urea nitrogen concentrations.

In the present study, the expression levels of the pluripotency factors SSEA-4, NANOG and OCT4 in induced PBMCs were detected by flow cytometry, and the results revealed that PBMCs dedifferentiated into multipotent cells. The quantitative PCR analysis demonstrated that the telomeres in induced PBMCs were significantly longer than those in noninduced PBMCs. Based on our evidence, PBMCs obtained from rabbit peripheral blood dedifferentiate into multipotent stem cells following treatment with chicken egg white extracts and exhibit the characteristics of mesenchymal stem cells [9], which allows their eventual participation in the repair of renal injury. Studies have shown that mammalian egg cell and Xenopus egg cell extracts can reprogram somatic cells [6, 7]. Our previous research showed that chicken egg extract can also reprogram somatic cells [5, 9]. The chicken egg is the largest egg cell, and its ability to reprogram somatic cells will thus advance cell biological research. The mammalian egg cell and Xenopus egg cell extract-induced reprogramming steps are cumbersome, and the extracts are difficult to obtain; however, chicken eggs are the largest egg cells, and a large amount of chicken egg white extract can thus be obtained. If the procedure is performed in a sterile manner, the obtained extract does not need to be filtered and sterilized. The activity of the extract can be maintained well. We repeated the induction experiment using final concentrations of chicken egg white extract of 10%, 20%, 30%, 40%, and 50% to induce the PBMCs (see S1 to S3 Figs). As the concentration of the chicken egg white extract increased, the pluripotency factor positive rate gradually increased, and the highest positive rate was obtained with a final concentration of 50%. However, a chicken egg white extract concentration higher than 50% will affect cell growth. We used protease, DNase, and RNase to lyse the protein, DNA, and RNA in egg white extract, respectively, and then conducted our induction experiment. We found that the egg white extract obtained after protein lysis no longer has the ability to reprogram cells, and the chicken protein extract obtained after DNA and RNA lysis still has the ability to reprogram cells. Therefore, these findings confirmed that the main role of the extract is played by the protein component (results not shown).

We also analyzed animal blood samples after one week of treatment. However, the difference among the four groups was not significant; thus, we only show the results obtained for the four groups after three treatments. The distribution of fluorescently labeled induced PBMCs in frozen kidney sections suggested that these cells were involved in repairing the injured kidney. HE staining showed damage to the kidney structure in the model control group. After treatment with induced cells, the kidney exhibited a normal structure, whereas damage to the kidney structure was observed in the noninduced cell treatment group. After three treatments, Masson's trichrome staining of kidney sections from the four groups revealed substantial collagen fiber deposition and serious fibrosis in the model control group. This fibrosis was improved or eliminated by treatment with the induced cells, whereas severe fibrosis was still observed in the noninduced cell treatment group. A renal metabonomics analysis of the four groups conducted in the negative ion mode revealed significantly increased levels of 2'-deoxy-D-ribose and N-acetylglucosamine 1-phosphate in the model control group compared with the normal control group, and the levels of these substances were significantly reduced in the induced cell treatment group compared with the model control group. In the positive ion mode, the levels of five substances were significantly increased in the model control group compared with the normal control group, and the levels of three of these five substances were significantly reduced in the induced cell treatment group compared with the model control group.

The postulated main mechanisms underlying the observed repair of the kidney and the accompanying reduction in the inflammatory response are listed below. ① Induced PBMCs differentiate into TECs or fuse with surviving cells to directly promote the repair of renal tissue. ② The synergistic effects of various cytokines provide a good microenvironment for the repair of renal tissues [26]. After three transplantations of induced PBMCs, the renal function and pathological indexes returned to the normal levels, and these normal levels persisted until the end of the experiment, which suggested that the transplanted multipotent stem cells effectively promoted the repair of renal tissue structure and function.

Although the specific mechanism through which induced PBMCs promote structural and functional repair of the kidney has not yet been fully understood, the results of this study suggest that the intravenous transplantation of induced PBMCs promotes repair after acute kidney injury. These results provide a valuable reference for researchers investigating the function of multipotent stem cells in regenerating damaged kidneys and for the clinical treatment of acute and chronic kidney diseases.

In summary, induced multipotent stem cell transplantation has substantial significance for parenchymal cell repair in animal models of renal failure and other kidney diseases.

## Conclusions

The treatment of PBMCs with chicken egg white extract significantly increased the expression of pluripotency-related genes and proteins, which indicated that the cells had dedifferentiated into multipotent stem cells. Thus, induced PBMCs dedifferentiate into multipotent stem cells and can potentially be used to treat kidney injury. Future research is needed to identify the key molecules in chicken egg white extract and thus further improve the induction efficiency.

## Supporting information

**S1 Fig.**
(TIF)

**S2 Fig.**
(TIF)

**S3 Fig.**
(PPTX)

## Acknowledgments

We thank American Journal Experts for assisting with the preparation of this manuscript.

## Author Contributions

**Data curation:** Xiang Yao, Qing-keng Lin, Zi-an Li.

**Formal analysis:** Xiang Yao.

**Funding acquisition:** Rong-qing Pang, Xing-hua Pan.

**Methodology:** Zi-an Li.

**Supervision:** Xue-min Cai.

**Validation:** Xue-min Cai, Xing-hua Pan.

**Visualization:** Xing-hua Pan.

**Writing – original draft:** Guang-ping Ruan.

**Writing – review & editing:** Guang-ping Ruan.

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
