## [Decision Letter · Decision Letter 0]

27 Aug 2020

PONE-D-20-22977

Transplantation of Chicken Egg-White Extract-Induced Rabbit PBMCs as Treatment for Renal Ischemia-Reperfusion Injury in Rabbits

PLOS ONE

Dear Dr. Ruan,

Thank you for submitting your manuscript to PLOS ONE. After careful consideration, we feel that it has merit but does not fully meet PLOS ONE’s publication criteria as it currently stands. Therefore, we invite you to submit a revised version of the manuscript that addresses the points raised during the review process.

During revision, please give proper consideration to all the comments received from the reviewer. In addition, please try to remain consistent while discussing your results and do not exaggerate it. For example, in this manuscript you have not shown any similarity between your induced cells and mesenchymal stem cells. Hence, avoid any irrelevant discussion that does not support your findings. Furthermore, claiming the induced cells have dedifferentiated into pluripotent cells is a kind of exaggeration. Please try to make your remarks softer and realistic.     

We look forward to receiving your revised manuscript.

Kind regards,

Nazmul Haque

Academic Editor

PLOS ONE

Journal Requirements:

2. In your Methods section, please provide additional details regarding the rabbits used in your study and ensure you have described the source. For more information regarding PLOS' policy on materials sharing and reporting, see https://journals.plos.org/plosone/s/materials-and-software-sharing#loc-sharing-materials.

3. In your Methods section, please state the volume of the blood samples collected for use in your study.

4.PLOS ONE now requires that authors provide the original uncropped and unadjusted images underlying all blot or gel results reported in a submission’s figures or Supporting Information files. This policy and the journal’s other requirements for blot/gel reporting and figure preparation are described in detail at https://journals.plos.org/plosone/s/figures#loc-blot-and-gel-reporting-requirements and https://journals.plos.org/plosone/s/figures#loc-preparing-figures-from-image-files. When you submit your revised manuscript, please ensure that your figures adhere fully to these guidelines and provide the original underlying images for all blot or gel data reported in your submission. See the following link for instructions on providing the original image data: https://journals.plos.org/plosone/s/figures#loc-original-images-for-blots-and-gels.

5. To comply with PLOS ONE submissions requirements, in your Methods section, please provide additional information on the animal research and ensure you have included details on efforts to alleviate animal suffering.

6.Thank you for stating the following in the Acknowledgments Section of your manuscript:

[This work was supported by grants from Yunnan Science and Technology Plan Project

Major Science and Technology Project (2018ZF007) and the Yunnan Province

Applied Basic Research Program Key Project (2018FA041, 2017FA040). National

Natural Science Foundation (31970515), 920th Hospital of the PLA Joint Logistics

Support Force In-hospital technology plans (2019YGB17, 2019YGA05).]

 [The funders had no role in study design, data collection and analysis, decision to publish, or preparation of the manuscript.]

7. Your ethics statement must appear in the Methods section of your manuscript. If your ethics statement is written in any section besides the Methods, please move it to the Methods section and delete it from any other section. Please also ensure that your ethics statement is included in your manuscript, as the ethics section of your online submission will not be published alongside your manuscript.

Reviewers' comments:

Reviewer's Responses to Questions

**Comments to the Author**

1. Is the manuscript technically sound, and do the data support the conclusions?

Reviewer #1: Yes

Reviewer #2: Partly

Reviewer #3: No

2. Has the statistical analysis been performed appropriately and rigorously? 

Reviewer #1: Yes

Reviewer #2: Yes

Reviewer #3: Yes

3. Have the authors made all data underlying the findings in their manuscript fully available?

Reviewer #1: Yes

Reviewer #2: No

Reviewer #3: No

4. Is the manuscript presented in an intelligible fashion and written in standard English?

Reviewer #1: Yes

Reviewer #2: Yes

Reviewer #3: No

5. Review Comments to the Author

Reviewer #1: The idea for this research is interesting and offers a numerous possibilities for future investigations such as clarification of strict molecular mechanism of beneficial induced PBMC treatment.

The basic question is how the authors decided to conduct just three procedures of induced PBMC administration. Why would not we suppose that perhaps one treatment is enough for restoration of kidney architecture and function? Why did not you analyze animal blood sample after first week of treatment? Please explain and the answers on these question incorporate in discussion. Additionally, restructure some part of discussion by comparing your results with other studies instead of simple paraphrase of already shown results.

Please, use uniform phrase ACUTE KIDNEY INJURY, not ACUTE RENAL FAILURE.

Reviewer #2: The current manuscript entitled ‘Transplantation of Chicken Egg-White Extract-Induced Rabbit PBMCs as Treatment for Renal Ischemia-Reperfusion Injury in Rabbits’ by RUAN et al. is moderately organised and represented. However the following points need to be addressed for betterment of their manuscript.

1# In the Materials and Methods section the author wrote that ‘As shown in our previous study, cells grown in 50% chicken egg-white extract culture medium will differentiate into pluripotent stem cells and may be used to treat disease.’ They need to cite their paper here.

2# To check the stemness property the author checked the expression level of Oct4, Nanog and Sox2. I wonder why they overlooked Klf4 and c-Myc detection. To characterize any induced pluripotent cell type OKSM expression is very important.

3# It is highly recommended to draw a diagram of this research where all treatment groups will be included for better understanding.

4# In the histopathological assessment, the author performed immunohistochemistry analysis of the fibrosis factor TGF-β only. Is it adequate to represent fibrosis by detecting only Tgf-b expression? The authors need to add couple of more markers.

5# In the result section, discussion under the heading of ‘Infusions of labeled PBMCs to treat the rabbit model’ is not sufficient. The author should write their findings very clearly in this section. It is highly recommended to summarize each result of every sub-category at the end of description.

6# In the result section, the author wrote ‘Statistical analysis showed that the two groups were statistically significant (n=3, p=0.003)’ regarding Figure 1D. However, I failed to find any indication of this significance in the figure!!! Furthermore, the author mentioned that the expression of the somatic cell gene LMNA was decreased. But Figure 1D does not show any difference among the two candidates.

7# For statistical analysis they mentioned n=3 in everywhere. Does it represent biological replicates or technical replicates? In case of quantitative PCR do they use biological replicates? Author need to mention it in their manuscript.

8# Author needs to change all gene symbols in Italic throughout the manuscript including Tables also.

9# The picture resolution of Figure 1E-1M is very poor. The numbers and description are very hard to read. Furthermore, authors used SSEA-4 candidate to show the status of pluripotency induction. But they did not show SSEA-4 transcript expression. It would be better if they synchronize their candidates for qPCR and Flow cytometric analysis.

10# In the methods section the author needs to add the name and company of instruments they used in this research. It is recommended to check the manuscript again and correct missing one.

11# In the Figure 2, the labelling of A-D is missing. Additionally, which one is noninduced and induced PBMCs? Authors need to be more careful to organise a manuscript.

12# In the Figure 2E, it is recommended to replace 1-5 numbering by adding text. And also mention candidates as usual.

13# In the Figure 2F, the picture resolution is very poor. Failed to read the figure text. There is no sign of statistically significance though author claim it in the manuscript.

14# In the Figure 2G-2I, I failed to read the text of it. Most probably, the statistical significance signs are also missing here.

15# The Figure 3A-3D is difficult to understand. The difference is not visible. Author needs to reedit their picture for better visualisation. Furthermore, 3A-3H figures have no identifying text!!!

16# The Figure 3I-3J is difficult to read. Most probably, the statistical significance signs are also missing here!!!

17# In the Figure 4, authors are asked to label A-H, to add statistical significance sign in I and to add text for magnification bar in J. Furthermore, the author wrote ‘Immunofluorescence results showed that the transplanted cells had green fluorescence and red fluorescence at the same time, indicating that the transplanted cells were differentiated into tubular epithelial cells.’ But it is very hard to find any red fluorescence in Figure 4J. Need to use arrow for any specific presentation.

18# In Figure 5, the author needs to add line bars among their comparable candidates in the figure. It is hardly detectable among which the statistical analysis was done.

19# In Figure 6, the author needs to edit all texts. Nothing is visible here. It is very tough to comment more than that….!!!

20# The author wrote in summary ‘the results of this study suggest that the intravenous transplantation of induced PBMCs promotes the repair of acute kidney injury’. However, they did not check any expression of marker genes. They need to add couple of marker gene expressions, if possible.

21# Authors need to submit their metabolomics data in any open access platform.

Reviewer #3: The manuscript, “Transplantation of Chicken Egg-White Extract-Induced Rabbit PBMCs as Treatment for Renal Ischemia-Reperfusion Injury in Rabbits” does not present any novel scientific concept. The study was not presented properly, the figures were not labelled and there are number of unclear images presented without paying due attention.

• Induction of PBMCs by chicken egg white extract is an old concept. Number of studies have been conducted two-three decades ago. Heterogeneous nature of the chicken egg white components and allergenic substances could affect the outcome.

• Authors need to explain the results of the previous studies with egg white extract on inducing cells in the introduction.

• Chicken egg white contains different growth factors and enzymes. Which component of chicken egg white are you interested in and expecting to induce cells?

• The research gap addressed by the study is not well defined.

• How many rabbits were used for harvesting peripheral blood? Please explain the procedure and mention how the animal well-being was maintained.

• What is the rationale of using 50% egg white extract medium? How was it formulated? Was the protocol standardized? If so, please cite the methodology.

• Instead of formulating egg white extract medium, authors could have used purified egg white lysozymes as done in other studies to avoid interferences with host biology.

• Methodology is not sub-sectioned well. Please put a title instead of using sentences and explain the methodology thereafter.

• Results section also needs proper headings rather than using sentences

• In the discussion, the results of the egg white extract induction mechanism were not discussed.

• Please mention other methods that have been used for induction, and compare the results of those methods with egg white extract method.

• Figures are not labelled. Therefore, referring to the figures is very difficult.

• The axes of the graphs are not labelled properly.

• There are number of unclear images presented without paying due attention.

• The manuscript needs revision for English language.

6. PLOS authors have the option to publish the peer review history of their article (what does this mean?). If published, this will include your full peer review and any attached files.

Reviewer #1: No

Reviewer #2: No

Reviewer #3: No

---

## [Author Response · Author response to Decision Letter 0]

20 Sep 2020

Dear Editor,

Thank you very much for your letter and advice. We have revised the paper and would like to resubmit it for your consideration. We have addressed the reviewers’ comments. This manuscript has been edited by American Journal Experts.

We hope that our revised manuscript is acceptable for publication, and we look forward to hearing from you at your earliest convenience.

Best wishes,

Guang-ping Ruan

---

## [Decision Letter · Decision Letter 1]

8 Oct 2020

PONE-D-20-22977R1

Transplantation of Chicken Egg-White Extract-Induced Rabbit PBMCs as a Treatment for Renal Ischemia-Reperfusion Injury in Rabbits

PLOS ONE

Dear Dr. Ruan,

Thank you for submitting your manuscript to PLOS ONE. After careful consideration, we feel that it has merit but does not fully meet PLOS ONE’s publication criteria as it currently stands. Therefore, we invite you to submit a revised version of the manuscript that addresses the points raised during the review process.

We look forward to receiving your revised manuscript.

Kind regards,

Nazmul Haque

Academic Editor

PLOS ONE

Reviewers' comments:

Reviewer's Responses to Questions

**Comments to the Author**

1. If the authors have adequately addressed your comments raised in a previous round of review and you feel that this manuscript is now acceptable for publication, you may indicate that here to bypass the “Comments to the Author” section, enter your conflict of interest statement in the “Confidential to Editor” section, and submit your "Accept" recommendation.

Reviewer #1: All comments have been addressed

Reviewer #3: (No Response)

2. Is the manuscript technically sound, and do the data support the conclusions?

Reviewer #1: Yes

Reviewer #3: No

3. Has the statistical analysis been performed appropriately and rigorously? 

Reviewer #1: Yes

Reviewer #3: Yes

4. Have the authors made all data underlying the findings in their manuscript fully available?

Reviewer #1: Yes

Reviewer #3: Yes

5. Is the manuscript presented in an intelligible fashion and written in standard English?

Reviewer #1: Yes

Reviewer #3: No

6. Review Comments to the Author

Reviewer #1: (No Response)

Reviewer #3: Although authors have revised the manuscript up to a certain degree, there are inadequately addressed concerns as mentioned below.

1. As a response to the reviewer's comments, authors have mentioned that the induction effect of the chicken egg white extract obtained according to their methodology is "stable", with our giving any supportive evidence. How do the authors justify this?

2. Authors have added only a single sentence on chicken egg white extract as the previous literature. There should be a more extensive literature review in the introduction in relation to this.

3. What is the protein in chicken egg white extract that you consider as the active factor?

4. Research gap needs to be logically stated in the introduction.

5. How was the protocol on 50% egg white extract medium standardised?

7. PLOS authors have the option to publish the peer review history of their article (what does this mean?). If published, this will include your full peer review and any attached files.

Reviewer #1: No

Reviewer #3: No

---

## [Author Response · Author response to Decision Letter 1]

21 Oct 2020

PONE-D-20-22977R1

Transplantation of Chicken Egg-White Extract-Induced Rabbit PBMCs as a Treatment for Renal Ischemia-Reperfusion Injury in Rabbits

PLOS ONE

Dear Dr. Ruan,

Thank you for submitting your manuscript to PLOS ONE. After careful consideration, we feel that it has merit but does not fully meet PLOS ONE’s publication criteria as it currently stands. Therefore, we invite you to submit a revised version of the manuscript that addresses the points raised during the review process.

We look forward to receiving your revised manuscript.

Kind regards,

Nazmul Haque

Academic Editor

PLOS ONE

Dear Editor,

Thank you very much for your letter and advice. We have revised the paper and would like to resubmit it for your consideration. We have addressed the reviewers’ comments. This manuscript has been edited by American Journal Experts.

We hope that our revised manuscript is acceptable for publication, and we look forward to hearing from you at your earliest convenience.

Best wishes,

Guang-ping Ruan

We would like to express our sincere thanks to the reviewers for their constructive and positive comments.

Reviewers' comments:

Reviewer's Responses to Questions

Comments to the Author

1. If the authors have adequately addressed your comments raised in a previous round of review and you feel that this manuscript is now acceptable for publication, you may indicate that here to bypass the “Comments to the Author” section, enter your conflict of interest statement in the “Confidential to Editor” section, and submit your "Accept" recommendation.

Reviewer #1: All comments have been addressed

Reviewer #3: (No Response)

2. Is the manuscript technically sound, and do the data support the conclusions?

Reviewer #1: Yes

Reviewer #3: No

3. Has the statistical analysis been performed appropriately and rigorously?

Reviewer #1: Yes

Reviewer #3: Yes

4. Have the authors made all data underlying the findings in their manuscript fully available?

Reviewer #1: Yes

Reviewer #3: Yes

5. Is the manuscript presented in an intelligible fashion and written in standard English?

Reviewer #1: Yes

Reviewer #3: No

6. Review Comments to the Author

Reviewer #1: (No Response)

Responses to Reviewer 3

Reviewer #3: Although authors have revised the manuscript up to a certain degree, there are inadequately addressed concerns as mentioned below.

1. As a response to the reviewer's comments, authors have mentioned that the induction effect of the chicken egg white extract obtained according to their methodology is "stable", with our giving any supportive evidence. How do the authors justify this?

We repeated the induction experiment and performed flow cytometry, and the results showed that our induction experiment was stable. The result is as follows.

 As the concentration of the egg white extract increases, the positive rate gradually increases, and the positive rate is the highest at a final concentration of 50%.

2. Authors have added only a single sentence on chicken egg white extract as the previous literature. There should be a more extensive literature review in the introduction in relation to this.

We have made a more extensive literature review in the introduction in relation to this. “In our laboratory, induced multipotent stem cells are generated by treating peripheral blood mononuclear cells (PBMCs) from rabbit peripheral blood with a homemade egg-white extract to reverse differentiation[5]. This is the first report to use chicken egg-white extract to induce PBMCs and treat rabbit renal ischemia-reperfusion injury. Studies have shown that extracts of mammalian oocytes[6] and Xenopus oocytes[7] have the potential to reprogram cells. The identification of egg extracts with the ability to maintain and enhance the survival and differentiation of cells will be widely useful in cellular biology research. Many studies have reported that animal egg extracts are able to induce the reprogramming of somatic cells[6, 8]. The chicken egg yolk is the largest egg cell, where the yolk membrane comprises the cell membrane, and the egg white and eggshell, which have nutritional and protective roles, are formed by oviduct secretions. Therefore, chicken egg white extract has the capacity to induce stemness in PBMCs.”

3. What is the protein in chicken egg white extract that you consider as the active factor?

Because we have done the following experiments, using protease, DNase, and RNase to respectively lyse the protein, DNA, and RNA in the egg white extract, and then conduct our induction experiment, and found that the egg white extract after protein lysis no longer has the ability to reprogram cells, and the chicken protein extract after DNA and RNA lysis still has the ability to reprogram cells. So we proved that the main role of the extract is the protein component.

4. Research gap needs to be logically stated in the introduction.

We have added a sentence in the introduction: “Further experiments need to find the key molecules in the chicken egg white extract in order to further improve the induction efficiency and promote this method.”

5. How was the protocol on 50% egg white extract medium standardised?

“We repeated the induction experiment, using 10%, 20%, 30%, 40%, 50% final concentration of chicken egg-white extract to induce cells. As the concentration of the chicken egg-white extract increased, the pluripotency factor positive rate gradually increased, and at the final concentration of 50% the positive rate is the highest. But if the concentration of chicken egg-white extract exceeds 50%, cell growth will be affected. Thus, we used 50% chicken egg-white extract-induced rabbit PBMCs as a treatment for kidney injury in the rabbit model.” The result is shown below.

7. PLOS authors have the option to publish the peer review history of their article (what does this mean?). If published, this will include your full peer review and any attached files.

Do you want your identity to be public for this peer review? For information about this choice, including consent withdrawal, please see our Privacy Policy.

Reviewer #1: No

Reviewer #3: No

---

## [Editor Report · Decision Letter 2]

26 Oct 2020

PONE-D-20-22977R2

Transplantation of Chicken Egg-White Extract-Induced Rabbit PBMCs as a Treatment for Renal Ischemia-Reperfusion Injury in Rabbits

PLOS ONE

Dear Dr. Ruan,

Thank you for submitting your manuscript to PLOS ONE. After careful consideration, we feel that it has merit but does not fully meet PLOS ONE’s publication criteria as it currently stands. Therefore, we invite you to submit a revised version of the manuscript that addresses the points raised during the review process.

Please provide proper consideration to the Editor's comments and revise the manuscript carefully.

We look forward to receiving your revised manuscript.

Kind regards,

Nazmul Haque

Academic Editor

PLOS ONE

Additional Editor Comments (if provided):

1. Please remove the following sentences from the Materials and Methods and add in the discussion appropriately with proper citation. Mention the name of the cell type on which the experiment was conducted.

“We repeated the induction experiment, using 10%, 20%, 30%, 40%, 50% final concentration of chicken egg-white extract to induce cells. As the concentration of the chicken egg-white extract increased, the pluripotency factor positive rate gradually increased, and at the final concentration of 50% the positive rate is the highest. But if the concentration of chicken egg-white extract exceeds 50%, cell growth will be affected.”

2. In answer to the question 3 you have written the following sentences:

"Because we have done the following experiments, using protease, DNase, and RNase to respectively lyse the protein, DNA, and RNA in the egg white extract, and then conduct our induction experiment, and found that the egg white extract after protein lysis no longer has the ability to reprogram cells, and the chicken protein extract after DNA and RNA lysis still has the ability to reprogram cells. So we proved that the main role of the extract is the protein component.”

Please add these sentences in the discussion appropriately with proper citation.

---

## [Author Response · Author response to Decision Letter 2]

26 Oct 2020

PONE-D-20-22977R2

Transplantation of Chicken Egg-White Extract-Induced Rabbit PBMCs as a Treatment for Renal Ischemia-Reperfusion Injury in Rabbits

PLOS ONE

Dear Dr. Ruan,

Thank you for submitting your manuscript to PLOS ONE. After careful consideration, we feel that it has merit but does not fully meet PLOS ONE’s publication criteria as it currently stands. Therefore, we invite you to submit a revised version of the manuscript that addresses the points raised during the review process.

Please provide proper consideration to the Editor's comments and revise the manuscript carefully.

We look forward to receiving your revised manuscript.

Kind regards,

Nazmul Haque

Academic Editor

PLOS ONE

Dear Editor,

Thank you very much for your letter and advice. We have revised the paper and would like to resubmit it for your consideration. We have addressed the reviewers’ comments. This manuscript has been edited by American Journal Experts.

We hope that our revised manuscript is acceptable for publication, and we look forward to hearing from you at your earliest convenience.

Best wishes,

Guang-ping Ruan

We would like to express our sincere thanks to the reviewers for their constructive and positive comments.

Additional Editor Comments (if provided):

1. Please remove the following sentences from the Materials and Methods and add in the discussion appropriately with proper citation. Mention the name of the cell type on which the experiment was conducted.

“We repeated the induction experiment, using 10%, 20%, 30%, 40%, 50% final concentration of chicken egg-white extract to induce cells. As the concentration of the chicken egg-white extract increased, the pluripotency factor positive rate gradually increased, and at the final concentration of 50% the positive rate is the highest. But if the concentration of chicken egg-white extract exceeds 50%, cell growth will be affected.”

We have removed the following sentences from the Materials and Methods and add in the discussion appropriately with proper citation. The name of the cell type on which the experiment was conducted was PBMCs. “We repeated the induction experiment, using 10%, 20%, 30%, 40%, 50% final concentration of chicken egg-white extract to induce PBMCs (see supplementary material). As the concentration of the chicken egg-white extract increased, the pluripotency factor positive rate gradually increased, and at the final concentration of 50% the positive rate is the highest. But if the concentration of chicken egg-white extract exceeds 50%, cell growth will be affected.”

2. In answer to the question 3 you have written the following sentences:

"Because we have done the following experiments, using protease, DNase, and RNase to respectively lyse the protein, DNA, and RNA in the egg white extract, and then conduct our induction experiment, and found that the egg white extract after protein lysis no longer has the ability to reprogram cells, and the chicken protein extract after DNA and RNA lysis still has the ability to reprogram cells. So we proved that the main role of the extract is the protein component.”

Please add these sentences in the discussion appropriately with proper citation.

We have added these sentences in the discussion appropriately. “Because we have done the following experiments, using protease, DNase, and RNase to respectively lyse the protein, DNA, and RNA in the egg white extract, and then conduct our induction experiment, and found that the egg white extract after protein lysis no longer has the ability to reprogram cells, and the chicken protein extract after DNA and RNA lysis still has the ability to reprogram cells. So we proved that the main role of the extract is the protein component (results are not displayed).”

---

## [Decision Letter · Decision Letter 3]

16 Nov 2020

PONE-D-20-22977R3

Transplantation of Chicken Egg-White Extract-Induced Rabbit PBMCs as a Treatment for Renal Ischemia-Reperfusion Injury in Rabbits

PLOS ONE

Dear Dr. Ruan,

Thank you for submitting your manuscript to PLOS ONE. After careful consideration, we feel that it has merit but does not fully meet PLOS ONE’s publication criteria as it currently stands. Therefore, we invite you to submit a revised version of the manuscript that addresses the points raised during the review process.

I appreciate your effort in addressing most of the issues raised by the reviewers. However, during this revision I would like to request you to provide proper consideration to the reviewers comments. Specially

1. Please revise the subtitles in the 'results' section and convert them into statements. 

2. Please revise the figures too, to make them more communicative. 

3. Provide proper explanation regarding data deposition.  

We look forward to receiving your revised manuscript.

Kind regards,

Nazmul Haque

Academic Editor

PLOS ONE

Reviewers' comments:

Reviewer's Responses to Questions

**Comments to the Author**

1. If the authors have adequately addressed your comments raised in a previous round of review and you feel that this manuscript is now acceptable for publication, you may indicate that here to bypass the “Comments to the Author” section, enter your conflict of interest statement in the “Confidential to Editor” section, and submit your "Accept" recommendation.

Reviewer #2: (No Response)

Reviewer #3: All comments have been addressed

2. Is the manuscript technically sound, and do the data support the conclusions?

Reviewer #2: Partly

Reviewer #3: Yes

3. Has the statistical analysis been performed appropriately and rigorously? 

Reviewer #2: Yes

Reviewer #3: Yes

4. Have the authors made all data underlying the findings in their manuscript fully available?

Reviewer #2: No

Reviewer #3: (No Response)

5. Is the manuscript presented in an intelligible fashion and written in standard English?

Reviewer #2: No

Reviewer #3: (No Response)

6. Review Comments to the Author

Reviewer #2: I would like to thank the editor for providing another opportunity to review the revised version of this current manuscript. I expected that there would be a significant change in the revised manuscript. However, it was not so. The following points are major issues to me –

The weakest part of this manuscript is the arrangement of their results. In the result section, they used 11 sub-headings to represent their results. Do any of the sub-headings represent a result? They used the experiment names only. Interpretation of result shows the merit of any research. The current format of this manuscript is very hard to understand for general readers like me.

Another drawback is the arrangement of figures. It seems to me that the authors are very reluctant to rearrange their figures. The texts of several figures are very small and faint. If I can not read it then how come I comment? The author answered that the figure will be clearer if the picture will open in illustration software. However, they can easily increase the size of their texts for better viewing. I am very sorry to say that the current formats of their Figures are not up to the mark. In Figure 3A-3D, it is very hard to detect the color. The author can easily make it visible by adjusting brightness of the picture. Anyway, it is author’s choice. But the point is that they must need to provide improve figures to think forward.

Data deposition is another major issue to me. The author was asked to deposit the metabolomics data to any open resource. They reply that they will deposit it. But when? After acceptance? They need to provide the accession number in this manuscript. However, the authors mentioned that all relevant data are within the manuscript, which is not true. This activity is against the PLoS guideline.

Reviewer #3: (No Response)

7. PLOS authors have the option to publish the peer review history of their article (what does this mean?). If published, this will include your full peer review and any attached files.

Reviewer #2: No

Reviewer #3: No

---

## [Author Response · Author response to Decision Letter 3]

24 Nov 2020

PONE-D-20-22977R3

Transplantation of Chicken Egg-White Extract-Induced Rabbit PBMCs as a Treatment for Renal Ischemia-Reperfusion Injury in Rabbits

PLOS ONE

Dear Dr. Ruan,

Thank you for submitting your manuscript to PLOS ONE. After careful consideration, we feel that it has merit but does not fully meet PLOS ONE’s publication criteria as it currently stands. Therefore, we invite you to submit a revised version of the manuscript that addresses the points raised during the review process.

I appreciate your effort in addressing most of the issues raised by the reviewers. However, during this revision I would like to request you to provide proper consideration to the reviewers comments. Specially

1. Please revise the subtitles in the 'results' section and convert them into statements.

We have revised the titles of the subsections in the 'Results' section and converted them into statements.

2. Please revise the figures too, to make them more communicative.

We have revised the figures to better convey the information.

3. Provide proper explanation regarding data deposition.

We have deposited the data in protocols.io to enhance the reproducibility of our results. The DOI link is dx.doi.org/10.17504/protocols.io.bpyrmpv6.

We have uploaded our figure files to the Preflight Analysis and Conversion Engine (PACE) digital diagnostic tool and ensured that the figures meet the PLOS requirements. 

We have deposited the data in protocols.io to enhance the reproducibility of our results. The DOI link is dx.doi.org/10.17504/protocols.io.bpyrmpv6.

We look forward to receiving your revised manuscript.

Kind regards,

Nazmul Haque

Academic Editor

PLOS ONE

Dear Editor,

Thank you very much for your letter and advice. We have revised the paper and would like to resubmit it for your consideration. We have addressed the reviewers’ comments, and the manuscript has been edited by American Journal Experts.

We hope that our revised manuscript is acceptable for publication and look forward to hearing from you at your earliest convenience.

Best wishes,

Guang-ping Ruan

We would like to express our sincere appreciation to the reviewers for their constructive and positive comments.

Reviewers' comments:

Reviewer's Responses to Questions

Comments to the Author

1. If the authors have adequately addressed your comments raised in a previous round of review and you feel that this manuscript is now acceptable for publication, you may indicate that here to bypass the “Comments to the Author” section, enter your conflict of interest statement in the “Confidential to Editor” section, and submit your "Accept" recommendation.

Reviewer #2: (No Response)

Reviewer #3: All comments have been addressed

2. Is the manuscript technically sound, and do the data support the conclusions?

Reviewer #2: Partly

Reviewer #3: Yes

3. Has the statistical analysis been performed appropriately and rigorously?

Reviewer #2: Yes

Reviewer #3: Yes

4. Have the authors made all data underlying the findings in their manuscript fully available?

Reviewer #2: No

Reviewer #3: (No Response)

5. Is the manuscript presented in an intelligible fashion and written in standard English?

Reviewer #2: No

Reviewer #3: (No Response)

The manuscript has been edited by American Journal Experts again.

6. Review Comments to the Author

Responses to Reviewer 2

Reviewer #2: I would like to thank the editor for providing another opportunity to review the revised version of this current manuscript. I expected that there would be a significant change in the revised manuscript. However, it was not so. The following points are major issues to me –

The weakest part of this manuscript is the arrangement of their results. In the result section, they used 11 sub-headings to represent their results. Do any of the sub-headings represent a result? They used the experiment names only. Interpretation of result shows the merit of any research. The current format of this manuscript is very hard to understand for general readers like me.

We have revised the description and interpretation of the results to show the merits of our research. The current format of this manuscript allows its easy comprehension by general readers.

Another drawback is the arrangement of figures. It seems to me that the authors are very reluctant to rearrange their figures. The texts of several figures are very small and faint. If I can not read it then how come I comment? The author answered that the figure will be clearer if the picture will open in illustration software. However, they can easily increase the size of their texts for better viewing. I am very sorry to say that the current formats of their Figures are not up to the mark. In Figure 3A-3D, it is very hard to detect the color. The author can easily make it visible by adjusting brightness of the picture. Anyway, it is author’s choice. But the point is that they must need to provide improve figures to think forward.

We have adjusted the brightness of Figs 3A-3D and uploaded Figs 3A-3D and Figs 1E-1M separately. We have uploaded our figure files to the Preflight Analysis and Conversion Engine (PACE) digital diagnostic tool and ensured that the figures meet the PLOS requirements.

Data deposition is another major issue to me. The author was asked to deposit the metabolomics data to any open resource. They reply that they will deposit it. But when? After acceptance? They need to provide the accession number in this manuscript. However, the authors mentioned that all relevant data are within the manuscript, which is not true. This activity is against the PLoS guideline.

We have deposited the data in protocols.io to enhance the reproducibility of our results. The DOI link is dx.doi.org/10.17504/protocols.io.bpyrmpv6.

Reviewer #3: (No Response)

7. PLOS authors have the option to publish the peer review history of their article (what does this mean?). If published, this will include your full peer review and any attached files.

Do you want your identity to be public for this peer review? For information about this choice, including consent withdrawal, please see our Privacy Policy.

Reviewer #2: No

Reviewer #3: No

We have uploaded our figure files to the Preflight Analysis and Conversion Engine (PACE) digital diagnostic tool and ensured that the figures meet the PLOS requirements.

---

## [Editor Report · Decision Letter 4]

27 Nov 2020

PONE-D-20-22977R4

Transplantation of Chicken Egg White Extract-Induced Rabbit PBMCs as a Treatment for Renal Ischemia-Reperfusion Injury in Rabbits

PLOS ONE

Dear Dr. Ruan,

Thank you for submitting your manuscript to PLOS ONE. After careful consideration, we feel that it has merit but does not fully meet PLOS ONE’s publication criteria as it currently stands. Therefore, we invite you to submit a revised version of the manuscript that addresses the points raised during the review process.

Specifically, please revise the subheadings of the results section according to the suggestions given below in the Additional Editor Comments (if provided) section.

We look forward to receiving your revised manuscript.

Kind regards,

Nazmul Haque

Academic Editor

PLOS ONE

Additional Editor Comments (if provided):

The authors have addressed almost all the issues. However, the following issues needed to be resolved before further consideration of this manuscript for publication.

1. Please revise the subheadings of the results section carefully. Few subheading are too long to consider as subheadings. Please revise the subheadings following the examples given below:

"TGF-β immunohistochemical analyses showed that the IOD of the induced cell groups was significantly reduced." change to "The IOD of the induced cell groups was significantly reduced:"

"Masson’s trichrome staining showed that fibrosis was improved in the induced group" change to "Fibrosis was improved in the induced group:"

"Renal PAS staining showed that the basement membrane did not display significant thickening in the induced group." change to "Thickening of the basement membrane was not observed in the induced group:"

"The results from renal tubular epithelial cell immunofluorescence showed that the transplanted cells were differentiated into tubular epithelial cells." change to "Transplanted cells differentiated into tubular epithelial cells:"

"The analysis of renal metabolomics pathways showed that the pyrimidine metabolism and phenylalanine, tyrosine and tryptophan biosynthesis pathways were significant." change to "Significant changes in the pyrimidine metabolism and phenylalanine, tyrosine and tryptophan biosynthesis pathways were observed"

---

## [Author Response · Author response to Decision Letter 4]

29 Nov 2020

PONE-D-20-22977R4

Transplantation of Chicken Egg White Extract-Induced Rabbit PBMCs as a Treatment for Renal Ischemia-Reperfusion Injury in Rabbits

PLOS ONE

Dear Dr. Ruan,

Thank you for submitting your manuscript to PLOS ONE. After careful consideration, we feel that it has merit but does not fully meet PLOS ONE’s publication criteria as it currently stands. Therefore, we invite you to submit a revised version of the manuscript that addresses the points raised during the review process.

Specifically, please revise the subheadings of the results section according to the suggestions given below in the Additional Editor Comments (if provided) section.

We have revised the subheadings of the results section according to the suggestions given below in the Additional Editor Comments (if provided) section.

We have deposited our laboratory protocols in protocols.io to enhance the reproducibility of our results. The DOI link is dx.doi.org/10.17504/protocols.io.bpyrmpv6.

We look forward to receiving your revised manuscript.

Kind regards,

Nazmul Haque

Academic Editor

PLOS ONE

Dear Editor,

Thank you very much for your letter and advice. We have revised the paper and would like to resubmit it for your consideration. We have addressed the reviewers’ comments, and the manuscript has been edited by American Journal Experts.

We hope that our revised manuscript is acceptable for publication and look forward to hearing from you at your earliest convenience.

Best wishes,

Guang-ping Ruan

Additional Editor Comments (if provided):

The authors have addressed almost all the issues. However, the following issues needed to be resolved before further consideration of this manuscript for publication.

1. Please revise the subheadings of the results section carefully. Few subheading are too long to consider as subheadings. Please revise the subheadings following the examples given below:

We have revised the subheadings of the results section carefully. We have revised the subheadings following the examples given below.

"TGF-β immunohistochemical analyses showed that the IOD of the induced cell groups was significantly reduced." change to "The IOD of the induced cell groups was significantly reduced:"

"Masson’s trichrome staining showed that fibrosis was improved in the induced group" change to "Fibrosis was improved in the induced group:"

"Renal PAS staining showed that the basement membrane did not display significant thickening in the induced group." change to "Thickening of the basement membrane was not observed in the induced group:"

"The results from renal tubular epithelial cell immunofluorescence showed that the transplanted cells were differentiated into tubular epithelial cells." change to "Transplanted cells differentiated into tubular epithelial cells:"

"The analysis of renal metabolomics pathways showed that the pyrimidine metabolism and phenylalanine, tyrosine and tryptophan biosynthesis pathways were significant." change to "Significant changes in the pyrimidine metabolism and phenylalanine, tyrosine and tryptophan biosynthesis pathways were observed"

We have uploaded our figure files to the Preflight Analysis and Conversion Engine (PACE) digital diagnostic tool and ensured that the figures meet the PLOS requirements.

---

## [Editor Report · Decision Letter 5]

4 Dec 2020

Transplantation of Chicken Egg White Extract-Induced Rabbit PBMCs as a Treatment for Renal Ischemia-Reperfusion Injury in Rabbits

PONE-D-20-22977R5

Dear Dr. Ruan,

We’re pleased to inform you that your manuscript has been judged scientifically suitable for publication and will be formally accepted for publication once it meets all outstanding technical requirements.

Kind regards,

Nazmul Haque

Academic Editor

PLOS ONE
---

## [Editor Report · Acceptance letter]

9 Dec 2020

PONE-D-20-22977R5 

Transplantation of Chicken Egg White Extract-Induced Rabbit PBMCs as a Treatment for Renal Ischemia-Reperfusion Injury in Rabbits 

Dear Dr. Ruan:

I'm pleased to inform you that your manuscript has been deemed suitable for publication in PLOS ONE. Congratulations! Your manuscript is now with our production department. 

Kind regards, 

on behalf of

Dr. Nazmul Haque 

Academic Editor

PLOS ONE